# In Silico Discovery of Small-Molecule Inhibitors Targeting SARS-CoV-2 Main Protease

**DOI:** 10.3390/molecules28145320

**Published:** 2023-07-10

**Authors:** Menghan Gao, Dongwei Kang, Na Liu, Yanna Liu

**Affiliations:** 1School of Pharmacy and Pharmaceutical Sciences & Institute of Materia Medica, Shandong First Medical University & Shandong Academy of Medical Sciences, 6699 Qingdao Road, Jinan 250117, China; 2NHC Key Laboratory of Biotechnology Drugs, Shandong Academy of Medical Sciences, 6699 Qingdao Road, Jinan 250117, China; 3Key Lab for Rare & Uncommon Diseases of Shandong Province, 6699 Qingdao Road, Jinan 250117, China; 4Department of Medicinal Chemistry, School of Pharmaceutical Sciences, Cheeloo College of Medicine, Shandong University, 44 West Culture Road, Jinan 250012, China

**Keywords:** SARS-CoV-2, main protease inhibitors, molecular docking-based virtual screening, MD simulation

## Abstract

The COVID-19 pandemic has caused severe health threat globally, and novel SARS-Cov-2 inhibitors are urgently needed for antiviral treatment. The main protease (M^pro^) of the virus is one of the most effective and conserved targets for anti-SARS-CoV-2 drug development. In this study, we utilized a molecular docking-based virtual screening approach against the conserved catalytic site to identify small-molecule inhibitors of SARS-CoV-2 M^pro^. Further biological evaluation helped us identify two compounds, AF-399/40713777 and AI-942/42301830, with moderate inhibitory activity. Besides that, the in silico data, including molecular dynamics (MD) simulation, binding free energy calculations, and AMDET profiles, suggested that these two hits could serve as the starting point for the future development of COVID-19 intervention treatments.

## 1. Introduction

The COVID-19 pandemic, caused by SARS-CoV-2, has been a worldwide health emergency since 2019. SARS-CoV-2, which belongs to the Coronaviridae family, has a single-stranded positive RNA genome. The infection of SARS-CoV-2 could cause pneumonia, fatigue, diarrhea, and life-threatening cardiovascular complications or multiorgan failure. Two antiviral drugs (Remdesivir [1] and Nirmatrelvir [2]) have been approved by the FDA for emergency use. But further clinical data have shown that Remdesivir only has weak potency against SARS-CoV-2 infection [3], and the application of Nirmatrelvir can only prevent severe symptoms of the infection [4]. Novel anti-SARS-CoV-2 agents are still urgently needed.

The replication of SARS-CoV-2 can be divided into several steps, including fusion and entry, translation, polyprotein processing, RNA replication, viral assembly, and particle release (Figure 1). All of these steps are catalyzed by various enzymes of the virus (protease, RNA-dependent RNA polymerase, etc.) [5,6]. Theoretically, the inhibition of any enzyme could stop the replication of the virus. The fusion and entry of SARS-CoV-2 rely on the recognition of viral spike (S) protein and the host ACE2 (angiotensin-converting enzyme 2) [7]. The S protein is assembled as a homotrimer and is inserted in multiple copies into the phospholipid membrane of the virion, which gives it a crown-like appearance. The S protein consists of two non-covalently associated subunits: the S1 subunit binds ACE2 and the S2 subunit anchors the S protein to the viral membrane. The S2 subunit also includes a fusion peptide and other machinery necessary to mediate membrane fusion upon the infection of a new host cell. The receptor-binding domain (RBD) of the S1 subunit flips to an up conformation and binds to ACE2 [8]. Once the viral particle has fused and entered the cell, the viral positive-sense, single-stranded RNA genome (+ssRNA) is then exposed and translated into its polyprotein. The polyprotein is then cleaved into 16 non-structural proteins by two cysteine proteases, including the main protease (M^pro^) and papain-like protease (PL^pro^) [9,10]. M^pro^ cleaves the majority of non-structural proteins, which is crucial for viral replication, making it an ideal target for anti-viral agents. The viral genomic replication starts from the synthesis of full-length negative-sense RNA genome, which is the template for further production of +ssRNA. The newly produced +ssRNA is then used for the translation of polyprotein or for the packaging of new viral particles. The RNA replication relies on the replication and transcription complex (RTC), which contains RNA-dependent RNA polymerase (RdRp) and its cofactors, nsp7 and nsp8 [11,12]. Remdesivir, developed by Gilead, is approved by the FDA as novel nucleoside inhibitor of RdRp [1,3,13]. However, Remdesivir shows poor potency in inhibiting viral replication. Further investigation illustrated that RdRp possesses an in *trans* backtracking proofreading mechanism, which makes the inhibition potency of Remdesivir decreases badly [14]. The assembly and particle release of SARS-CoV-2 are also mediated by several enzymes from the virus and the host cell. The structural proteins (S protein, envelope (E) protein, membrane (M) protein, and nucleocapsid (N) protein) are translated, and the viral particle is assembled at the endoplasmic reticulum-to-Golgi compartment (ERGIC), where the viral RNA genome is encapsulated by the N protein and interacts with the M protein [15]. The M protein forms a scaffold for the virion and recruits the E and S proteins to the budding site. The E protein is a small membrane protein that regulates the membrane curvature and stability of the virion [16]. The S protein is a large glycoprotein that mediates the attachment and entry of the virus into the host cell by binding to the ACE2 receptor. The S protein undergoes proteolytic cleavage and conformational changes to expose its fusion peptide and facilitate membrane fusion. After assembly, the mature viral particles are transported from the ERGIC to the plasma membrane via vesicles. These vesicles fuse with the host cell membrane and release the virions via exocytosis [5,17].

M^pro^ is one of the most important enzymes that processes viral polyprotein into functional proteins. M^pro^ is a homodimer, where each monomer can be divided into two domains, the functional catalytic domain and the crucial dimerization domain. The catalytic site of M^pro^ consists of Cys145 and His41. The whole binding pocket can be identified as several sub-pockets (S1, S1’, S2, and S4). Occupation of the catalytic site could inhibit the normal function of M^pro^ and, thus, inhibit the replication of virus. The endogenous binding of ligands and the vital roles of M^pro^ make it an ideal target for drug design and development [17]. A shown in Figure 2, the inhibitor reported by Ma et al. [18] binds the catalytic site of M^pro^ and forms multiple hydrogen bonds with surrounding residues (Glu166, His163, Ser144, and Gly143). It also forms π-π stacking with His 41. The cocrystal structure of the M^pro^–inhibitor complex can shed light on the development of novel M^pro^ inhibitors.

To date, several ligands targeting M^pro^ have been reported (Figure 3) [19,20,21]. Nirmatrelvir, developed by Pfizer, acts as an orally active ligand in combination with the CYP3A inhibitor Ritonavir [4,22]. **GC-376** [23], **PF-00835231** [24,25], and **11a** [26] are currently under clinical trials. All of these peptidic inhibitors simulate the endogenous substrate of M^pro^ and possess warheads which could covalently bind with the vital catalytic residue Cys145.

Despite the high potency of peptidic covalent M^pro^ inhibitors, many of them show off-target side effects on host proteins. **GC-376** shows inhibitory activity toward several cathepsins (IC_50_ = 990, 74, and 0.56 nM to cathepsin L, cathepsin I, and cathepsin K, respectively). **PF-00835231** also shows inhibitory activity toward several cathepsins (IC_50_ = 146 nM and 1300 μM to cathepsin L and cathepsin B, respectively). **11a** is potent inhibitor of cathepsin L (IC_50_ = 210 nM) [18].

Compared to peptidic covalent M^pro^ inhibitors, non-peptidic inhibitors have attracted researchers’ attention due to their advantages, such as low off-target activity, higher oral bioavailability, and low toxicity. The non-peptidic inhibitors, **21**, **23R**, **S-216722**, and **Wu04**, display high antiviral activity and low toxicity (Figure 4) [27,28,29,30]. All of these findings suggest that non-peptidic M^pro^ inhibitors can be great potential candidate drugs for treating SARS-CoV-2 infection.

Computer-aided drug design (CADD) has been a general and practical technique highly used in the development of novel drugs; it has become a fundamental tool in the development of novel drugs for most pharmaceutical companies [31,32]. Several blockbuster drugs (Glivec, etc.) were developed using CADD [33]. In this context, virtual screening has been used to discover novel chemotypes of SARS-CoV-2 M^pro^ inhibitors. Approaches based on docking virtual screening using co-crystal structures, pharmacophore models, and quantitative structure–activity relationship (QSAR)-based screening are used to identify novel hits and guide the structure optimization of M^pro^ inhibitors. Moreover, molecular dynamics simulations have been exploited to investigate the binding mechanisms of these inhibitors [34,35].

Here, we report 17 compounds identified from molecular docking-based virtual screening targeting M^pro^. The initial biological evaluations identified two novel scaffolds that possessed moderate potency toward M^pro^. Molecular dynamics (MD) simulations of representative ligands targeting M^pro^ were performed to investigate the binding conformations of these ligands. The ADMET profiles of these ligands were also estimated to predict the potential draggability of representative molecules.

## 2. Results and Discussion

### 2.1. Molecular Docking-Based Virtual Screening

The cocrystal structure of M^pro^ with **23R** (PDB ID: 7KX5) was selected as the structure complex since few cocrystal structures had been reported when this work started. The binding pocket was identified using the binding pose of **23R**. Two commercial compound libraries (Enamine and Specs, total compound number > 3.5 M) were used for selection. A 3-step molecular docking-based virtual screening and a binding free energy (Molecular Mechanics/Generalized Born Surface Area, MM/GBSA) calculation were carried out. Finally, 17 compounds were selected based on the docking scores, MM/GBSA estimation, and manual selection (Figure 5, Table 1). The manual selection was based on the criteria reported by Fischer et al. [36], including fundamental interaction with key residues, binding conformation, and structural novelty.

### 2.2. Initial Biological Evaluation

These selected candidates were then purchased, and their enzyme inhibitory activities were evaluated using a fluorescence resonance energy transfer (FRET) assay (Table 2). Notably, AF-399/40713777 and AI-942/42301830 possess 47.11% and 37.67% of inhibitory activity toward M^pro^ at 100 μM, respectively. Moreover, AN-329/15538195, AN-655/14907067, AG-690/13705944, AK-968/37129380, and Z929753284 show >10% inhibition rate toward M^pro^. These encouraging biological results indicate these diverse scaffolds can be used for further SAR studies. These novel scaffolds provide more possibility for developing good candidates for treating SARS-CoV-2 infection.

### 2.3. Molecular Dynamics (MD) Simulation

MD simulation is a powerful computational technique that can provide insights into the structure, dynamics, and interactions of biomolecules at the atomic level. The initial biological results suggest that AF-399/40713777 and AI-942/42301830 possess moderate inhibitory activity toward SARS-CoV-2 M^pro^. The binding conformations of these two ligands with the pocket were then investigated. The MD simulation results show that both AF-399/40713777 and AI-942/42301830 bind to the catalytic site of M^pro^. The RMSD plots of the ligands heavy atoms and protein backbones were generated (Figure 6). The RMSF values of the ligands were also estimated (Figure 6). The computer data indicate that both of AF-399/40713777 and AI-942/42301830 have stable binding conformation with M^pro^, as shown through the 500 ns MD simulation.

The protein–ligand interaction diagram was further analyzed (Figure 7). AF-399/40713777 forms multiple interactions with surrounding residues. The carbonyl motifs interact with Gln189, Gln192, and Thr190 via hydrogen bond. The methoxy group forms hydrogen bond with Arg188. Besides these interactions, the phenyl ring forms π-π stacking with His41. These multiple interactions stabilize the binding conformation between the ligand and the pocket. 

AI-942/42301830 mainly binds to M^pro^ through abundant hydrophobic interactions. His41 forms multiple π-π stacking with the two phenyl rings of the ligand. It also interacts with Met49, Leu167, and Pro168 via hydrophobic interactions. Moreover, the carbonyl motif of ester forms hydrogen bond with Glu166.

### 2.4. Binding Free Energy (MM/GBSA) Calculations

The binding free energy (MM/GBSA) [37] of AF-399/40713777 (−66.07 kcal/mol) and AI-942/42301830 (−78.32 kcal/mol) with SARS-CoV-2 M^pro^ were estimated. The calculated binding affinities did translate well into experimentally determined inhibitory potencies (Table 3). As illustrated in Figure 7, AF-399/40713777 forms multiple hydrogen bonds with surrounding residues. And the ΔG_Hbond value of AF-399/40713777 is relatively higher compared to that of AI-942/42301830. On the other hand, AI-942/42301830 forms abundant hydrophobic interactions with the binding pocket. The ΔG_Lipo of AI-942/42301830 contributes more compared to that of AF-399/40713777.

### 2.5. Absorption, Distribution, Metabolism, Excretion, and Toxicity (ADMET) Prediction

ADMET is an important aspect of drug discovery and development as it determines the efficacy, safety, and optimal dosage of a drug. A poor ADMET profile is one of the main reasons for drug failure in clinical trials, resulting in wasted time, resources, and money. In order to eliminate compounds with undesirable pharmacokinetics and toxicity and reduce the risk of new drug discovery, we also evaluated the AMDET profiles of these two promising hits, AF-399/40713777 and AI-942/42301830, using ADMETlab 2.0 (Figure 8) and the ProTox-II server (Figure 9) [38,39]. ADMETlab 2.0 [38] is a web-based platform that provides comprehensive and accurate predictions of ADMET properties of molecules. ADMETlab 2.0 enables users to evaluate 17 physicochemical properties, 13 medicinal chemistry measures, 23 ADME endpoints, 27 toxicity endpoints, and 8 toxicophore rules of input molecules, using a multi-task graph attention framework. It also supports batch evaluation of molecular datasets and provides detailed explanation and optimal range of each property. ProTox-II [39] is a web server that provides predictions of oral toxicity for small molecules. It incorporates new features and functionalities, such as a more comprehensive database of toxic compounds, a more accurate prediction model, a more user-friendly interface, and a more informative output. ProTox-II aims to assist researchers and practitioners in the fields of drug discovery, chemical safety assessment, and environmental toxicology by providing reliable and interpretable toxicity predictions.

The prediction results showed that AF-399/40713777 has a desirable ADME profile, with proper logP and LogD, and acceptable LogS. AI-942/42301830 shows a relatively worse ADME profile, mainly because of the five phenyl rings in its structure. But both of them show desirable toxicity data. The predicted LD_50_ of AF-399/40713777 and AI-942/42301830 is 1000 mg/kg and 2000 mg/kg, respectively.

## 3. Materials and Methods

### 3.1. Target Protein Structure and Ligand Preparation

The cocrystal complex structure of M^pro^ with **23R** (PDB ID: 7KX5) obtained from a protein databank (rcsb.org) was used as the receptor structure. The protein preparation wizard, Schrödinger Suite 2022-2, was used to prepare the structure. Co-crystalized metals and ions and non-water solvents were deleted. The bond orders were reassigned using the Chemical Component Dictionary. The protein’s hydrogens were deleted and re-added. The terminal oxygen atoms were added, and water molecules beyond 8 Å of the binding ligand were deleted. The missing side chains of the whole structure were added and the hydrogen bonds were reassigned using PROPKA [40]. The whole structure was then energy minimized using the OPLS4 force field. The other parameters were set as default. 

More than 3 million commercially available compounds from two vendors (Specs and Enamine) were used for screening. All ligands were prepared using the Ligprep module. The ionization state of all ligands was optimized with pH = 7.0 using Epik, and alternative stereoisomers were determined from the original 3D structures. The other parameters were set as default.

### 3.2. Molecular Docking-Based Virtual Screening

The binding pocket was identified using the receptor grid generation module. The cocrystal complex structure of M^pro^ with **23R** (PDB ID: 7KX5) obtained from the protein databank (rcsb.org) was used to identify the binding pocket. All other parameters were set as default. 

The virtual screening was carried out using the virtual screening workflow of Schrödinger Suite 2022-2. All prepared compound libraries were used for the screening. The binding site was previously generated using the receptor grid generation module of Schrödinger Suite 2022-2, which was identified using the position of **23R**. The other parameters were set as default. With the binding grid box generated, Glide HTVS (high-throughput virtual screening) docking precision, standard precision (SP), and extra precision (XP) docking were used for screening in sequence. And the top 10% of best-scoring ligands for each procedure were selected for the next screening step. Eventually, the 10% best-scoring ligands (3208 ligands) after the extra precision (XP) docking were identified. The binding free energy (Molecular Mechanics/Generalized Born Surface Area, MM/GBSA) of the top 1000 ranking ligands with the receptor was estimated using the prime MMGBSA module [37]. The residues within 5 Å of the ligands were set as flexible, and all other parameters were set as default. The binding poses of these ligands were further checked manually. And the final candidates were selected based on the docking scores, binding free energy estimation, and manual selection (Table 1).

### 3.3. Initial Biological Evaluation of Selected Analogues

The enzyme inhibitory activity of selected candidates was evaluated using a previously reported well-developed fluorescence resonance energy transfer (FRET) assay [41].

A fluorescence resonance energy transfer (FRET) method was applied to measure M^pro^ inhibition of the tested compounds. Black 96-well plates, SARS-CoV-2 main protease solution (1 mg/mL, 500 mM of Tris, 150 mM of NaCl, 1 mM of EDTA, and 50% glycerol), and fluorescent substrate MCA-AVLQSGFR-Lys(Dnp)-Lys-NH_2_ (20 mM in DMSO) were purchased from Beyotime Biotechnology (Shanghai, China). M^pro^ was diluted to 1.5 μM using an assay buffer (50 mM of Tris-HCl, 150 mM of NaCl, 20% glycerol, and pH = 7.3) and preserved at −20 °C. The fluorescent substrate was diluted to 500 μM in an assay buffer (50 mM of Tris-HCl, 150 mM of NaCl, 1 mM of EDTA, and pH = 7.3). Stock solutions (5 mM) of the test compounds were prepared in DMSO and diluted to the desired concentration using the assay buffer. Each well of a black 96-well plate was added 50 μL of assay buffer, 20 μL of diluted enzyme solution, and 20 μL of inhibitor solution. The control wells (no inhibitor) and blank wells (no enzyme) were measured in parallel experiments. Then, the 96-well plate was incubated at 37 °C in a shaking incubator for 10 min. After incubation, 10 µL of substrate solution was added per well to initiate the reaction. At the excitation wavelength of 320 nM and the emission wavelength of 405 nM, the fluorescence signal was measured every 10 s for 10 min using a SpectraMax iD5 multimode plate reader (Molecular Devices). Data from the first 60 s were linear fitted to calculate the slope values (V_0_ for the control wells, and V_i_ for the test wells). %Inhibition (i%) in each well can, thus, be calculated using the following equation:i% = 1 − V_i_/V_0_ × 100%

### 3.4. Molecular Dynamics Simulations for Interaction Analyses and Binding Free Energy Estimations

The binding poses of AF-399/40713777 and AI-942/42301830 with M^pro^ were investigated using molecular dynamics simulation. The initial docking poses of these two ligands were used to build the simulation system. The protein–ligand complex within the explicit solvent system with the OPLS4 force field was studied using the Desmond module of Schrödinger suite 2022-1. The atomic framework was solvated with a TIP3P water model with orthorhombic intermittent limit conditions for a 10 Å buffer region. The overlapping water molecules were eliminated, and Na^+^ or Cl^−^ was added as counter ions to neutralize the entire framework of atoms. An extra 0.15 M of NaCl was added into the system. The simulation was performed using an ensemble (NPT) of Nosé–Hoover thermostat and barostat to maintain a constant temperature of 300 K and a pressure of 1 bar in the system (Maestro, Schrödinger suits 2022-1). A hybrid energy minimization algorithm with 1000 steps of steepest descent, followed by conjugate gradient algorithms, was used. Then, 500 ns molecular dynamics simulations (Maestro, Schrödinger suits 2022-1) were performed and the post-dynamics simulation was analyzed using the simulation interaction diagram module. The binding free energy was calculated using the MM/GBSA module of Maestro based on the molecular dynamics simulations. The whole simulation time (500 ns) was sampled. The MM/GBSA calculations were estimated every 5 ns. The average of all estimations is presented as the results. The detailed data can be found in the Appendix A.

### 3.5. ADMET Analysis

The ADMET profile of AF-399/40713777 and AI-942/42301830 were estimated using ADMET lab 2.0 (https://admetmesh.scbdd.com/ (accessed on 15 January 2023)). The molecular SIMLE strings were generated using Chemdraw and then submitted to the online system. An analysis report was then generated. The toxicity profile was evaluated using the ProTox-II server (https://tox-new.charite.de/ (accessed on 15 January 2023)).

## 4. Conclusions

In conclusion, we identified two novel non-peptidic inhibitors, AF-399/40713777 and AI-942/42301830, for SARS-CoV-2 M^pro^ using a molecular docking-based virtual screening approach. The initial enzyme inhibition assay indicated moderate inhibitory activities of both ligands toward M^pro^. The molecular dynamics simulations investigated the possible binding poses of these two ligands. In addition, the calculated binding affinities (MM/GBSA) did translate well into experimentally determined inhibitory potencies. The predicted ADMET results showed that AF-399/40713777 possessed a desirable profile. The successfully identified hits suggest the feasibility of this hierarchical virtual screening approach in drug discovery. Moreover, these two hits provide novel scaffolds for exploring high-activity inhibitors targeting SARS-CoV-2 M^pro^.

## Figures and Tables

**Figure 1 molecules-28-05320-f001:**
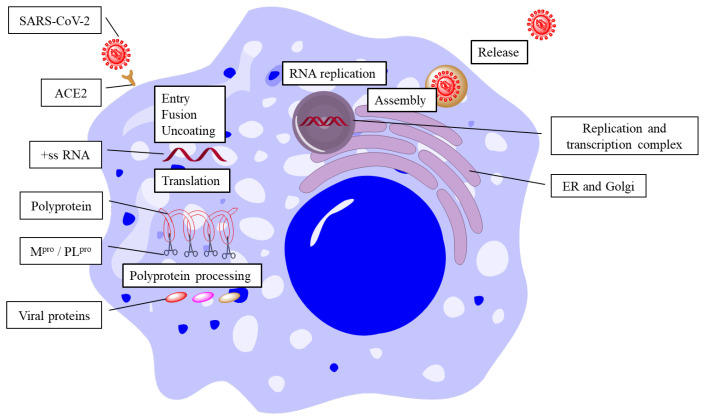
Life cycle of SARS-CoV-2.

**Figure 2 molecules-28-05320-f002:**
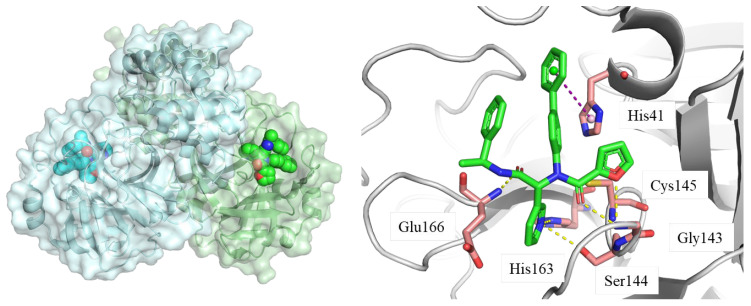
Crystal structure of M^pro^ homodimer and the catalytic site. Ligand: green stick/sphere; key residues: pink stick; hydrogen bond: yellow dashed line; π-π stacking: purple dashed line; PDB code: 7KX5 [18].

**Figure 3 molecules-28-05320-f003:**
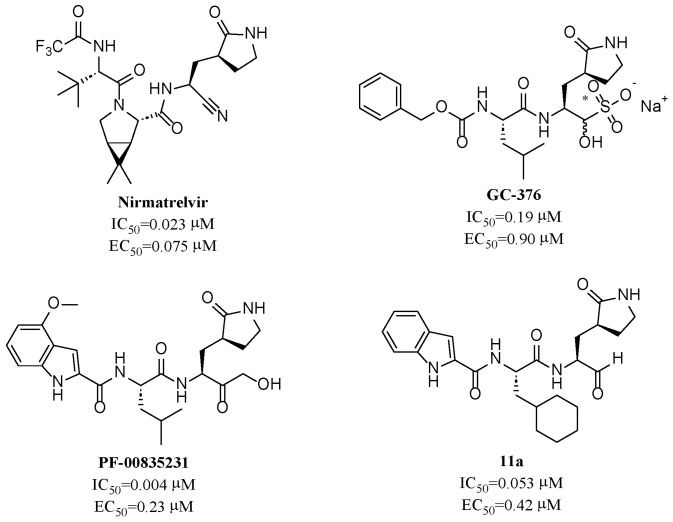
Representative peptidic covalent M^pro^ inhibitors.

**Figure 4 molecules-28-05320-f004:**
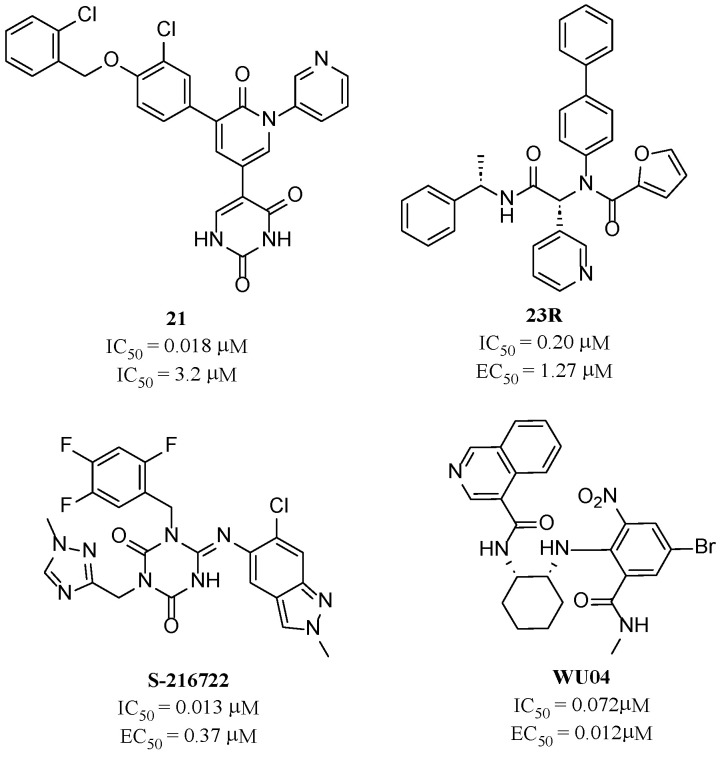
Representative non-peptidic M^pro^ inhibitors.

**Figure 5 molecules-28-05320-f005:**
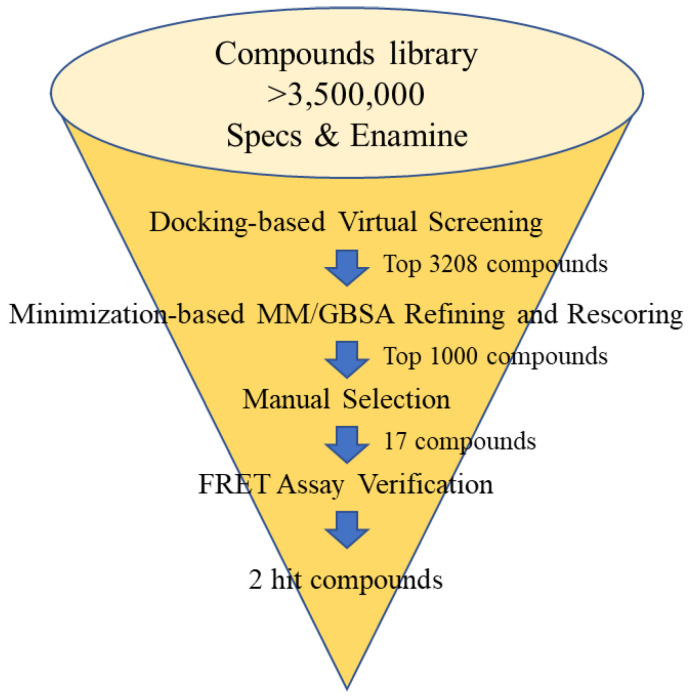
The flow chart of the discovery of inhibitors against SARS-CoV-2 M^pro^ via docking-based virtual screening.

**Figure 6 molecules-28-05320-f006:**
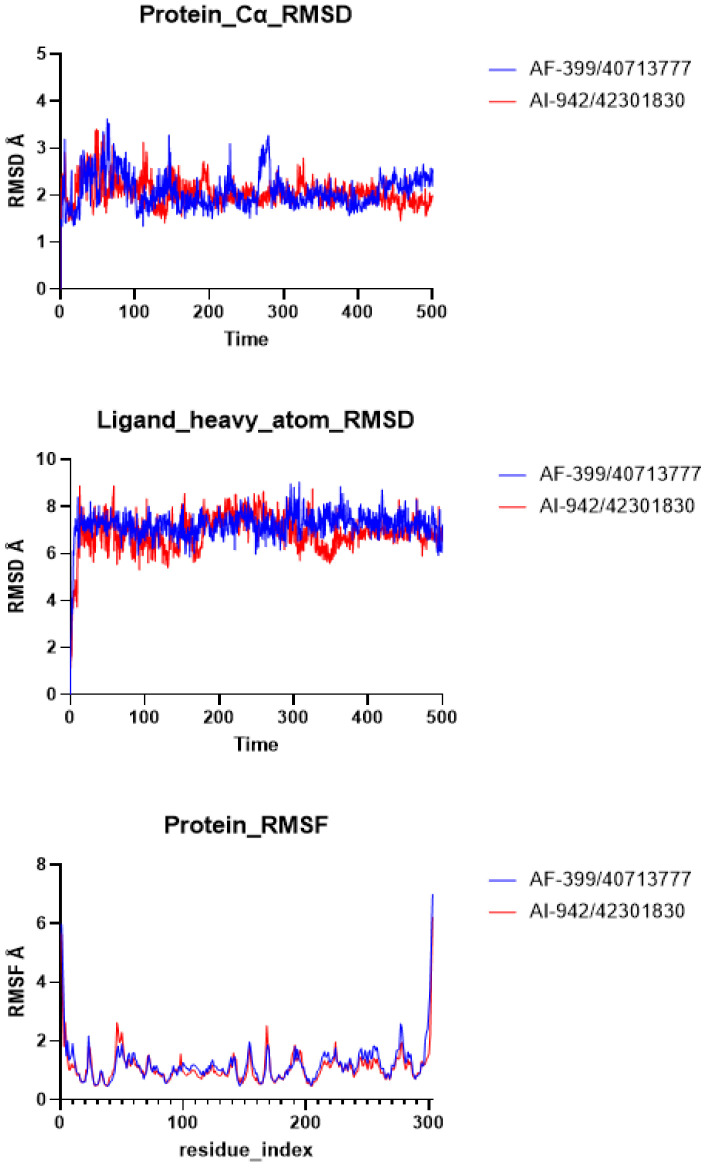
The RMSD and RMSF plots of AF-399/40713777 and AI-942/42301830 binding to M^pro^ generated via MD simulation.

**Figure 7 molecules-28-05320-f007:**
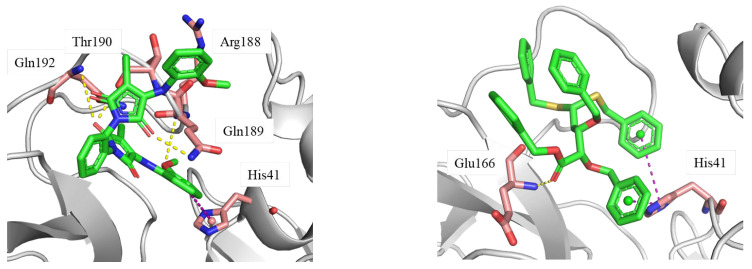
The protein–ligand binding conformations generated from the molecular dynamics simulation trajectory. Ligand: green stick; key residues: pink stick; hydrogen bond: yellow dashed line; π-π stacking: purple dashed line. The structure files (PDB files) can be found in the Appendix A.

**Figure 8 molecules-28-05320-f008:**
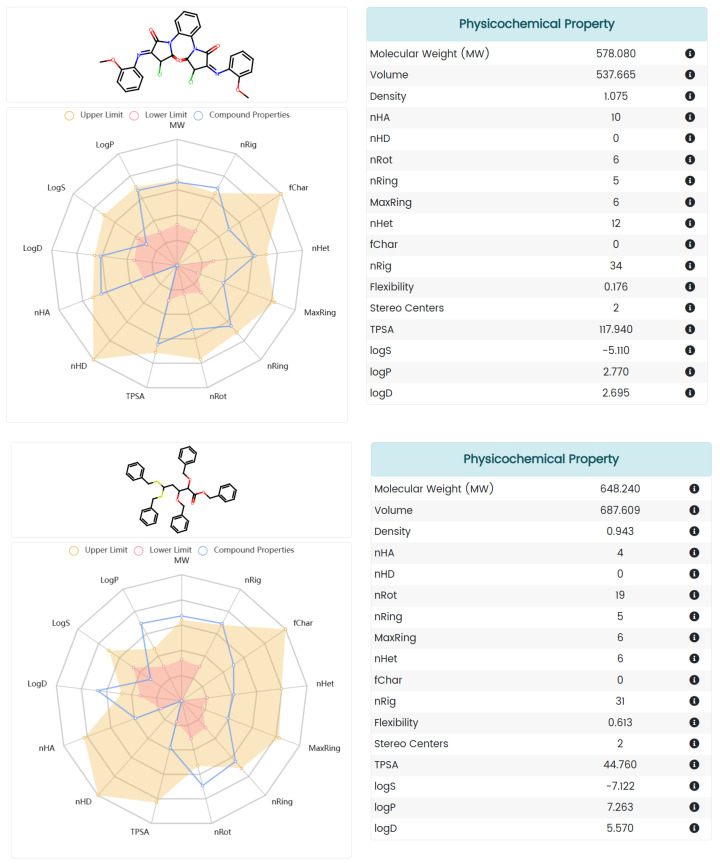
The ADMET profile prediction of small molecules processed using ADMET lab 2.0.

**Figure 9 molecules-28-05320-f009:**
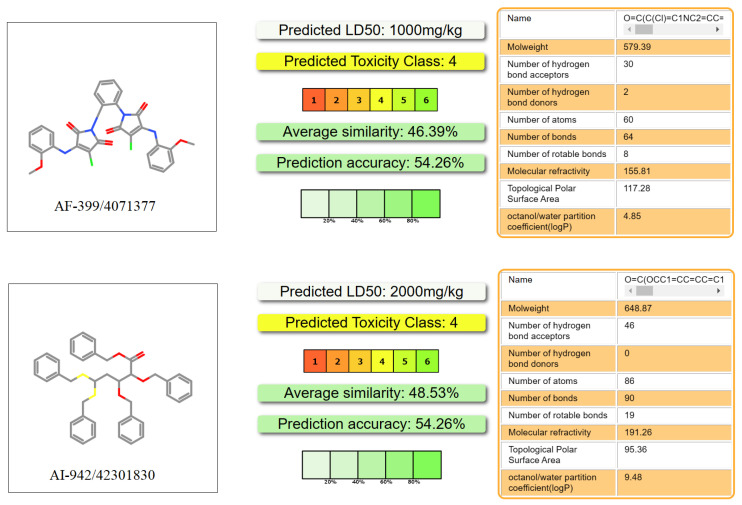
The toxicity prediction of representative ligands processed using the ProTox-II server.

**Table 1 molecules-28-05320-t001:** Docking scores and MM/GBSA (kcal/mol) of selected ligands as anti-SARS-CoV-2 M^pro^ candidates.

ID	Structure	Docking Score	MM/GBSA
AN-329/15538195	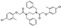	−7.769	−102.03
AF-399/40713777	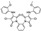	−7.929	−91.41
AN-655/14907067	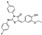	−7.882	−86.30
AK-968/12101028	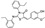	−7.911	−86.00
AG-690/13705944	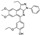	−8.568	−85.86
AK-968/37129380	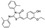	−7.829	−85.48
AG-205/36953218	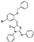	−8.248	−85.06
AH-487/11927009	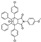	−7.764	−84.29
AI-942/42301830	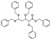	−8.856	−87.32
AN-329/14726055	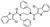	−8.798	−88.09
Z54217235	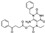	−8.670	−87.01
Z91218686	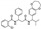	−9.095	−86.00
Z20007584	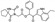	−10.038	−64.14
Z92376193	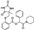	−9.139	−71.91
Z929753284	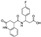	−9.018	−49.50
Z245966642	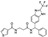	−8.850	−61.06
Z1603682175	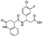	−8.827	−51.59

**Table 2 molecules-28-05320-t002:** Enzyme inhibitory activities of selected candidates.

ID	% Inhibition @ 100 μM	% Inhibition @ 100 μM Average of 3 Samples
AN-329/15538195	18.9%	11.80% ± 5.65
AF-399/40713777	23.7%	47.11% ± 0.84
AN-655/14907067	13.3%	2.71% ± 7.02
AK-968/12101028	4.3%	
AG-690/13705944	12.4%	
AK-968/37129380	12.5%	
AG-205/36953218	−13.8%	
AH-487/11927009	−0.1%	
AI-942/42301830	17.8%	37.67% ± 2.61
AN-329/14726055	−5.1%	−0.43% ± 3.97
Z54217235	−14.8%	
Z91218686	−6.1%	
Z20007584	4.5%	
Z92376193	−5.0%	
Z929753284	18.4%	
Z245966642	3.8%	
Z1603682175	7.2%	

**Table 3 molecules-28-05320-t003:** MM/GBSA (kcal/mol) of AF-399/40713777 and AI-942/42301830 with SARS-CoV-2 M^pro 1^.

Ligands	ΔG_Bind_	ΔG_Coulomb_	ΔG_Hbond_	ΔG_Lipo_	ΔG_vdW_
AF-399/40713777	−66.07	−15.60	−0.66	−16.14	−51.01
AI-942/42301830	−78.32	−11.98	−0.51	−25.66	−62.19

^1^ The whole simulations trajectory was sampled, and the estimation was calculated every 5 ns. The averages of all calculations are presented as the results. Complete data and standard deviations can be found in the Appendix A.

## Data Availability

The original contributions presented in the study are included in the article/Appendix A; further inquiries can be directed to the corresponding authors.

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
