# Peer review of "In Silico Discovery of Small-Molecule Inhibitors Targeting SARS-CoV-2 Main Protease"

_molecules, 2023, doi:10.3390/molecules28145320_

Round 1
Reviewer 1 Report
Appropriately done project showing virtual screening that lead to actual testing of these compounds on the target (Mpro). Main test for validation was a FRET based assay. No graphic/table/figure for those results given, just minimal verbiage in the text.
How did all compounds score with FRET?
Need to see real results to take this seriously.
Only 13 references seemed very sparse given so much in this field of topic in last 3 years.
Could use professional help to clean it up. Jumps in logic are hard to follow.
Author Response
The authors appreciate the comments from the reviewer, which can greatly improve the quality of the manuscript. Here is the response to the comments.
How did all compounds score with FRET?
Need to see real results to take this seriously.
The raw data of FRET experiment was provided as supplementary information.
Only 13 references seemed very sparse given so much in this field of topic in last 3 years.
More references were added.
Reviewer 2 Report
The article “In Silico Discovery of Small Molecule Inhibitors Targeting 2 SARS-Cov-2 Main Protease” presented by Menghan Gao et al addresses a very interesting topic with interesting results but fail to present it appropriately.
1- The introduction is very thorough in the explanation of the life cycle of the virus, but could be improved by the addition of a schematic representation of that cycle.
2- The introduction fails to correctly explain the catalytic mechanism of the Mpro. Line 88 in page 2 states that “could covalently bind with the vital catalyze residue His41”. Residue His41 does not carry out the nucleophilic attack on the substrate or covalent inhibitors. His41 activates Cys145 and it is Cys145 that carries out the nucleophilic attack.
3- The introduction fails to show the main point of interest of this article which is the structure of the Mpro and its interaction with bound inhibitors. This is crucial, as further along the text authors state in line 166 that they “identified two novel scaffolds” and in lines 249 “The binding pocket was identified”; but they have not provided any information whatsoever on the structure of the ligand-binding site of the known inhibitors, so the reader must take their word for it. Authors should at least use the crystal structure of Mpro bound to a ligand that they have used as starting structure for the computational experiments to explain the 3D structure of Mpro, the ligand-binding site and the key ligand-receptor interactions to be able to support their results further along the text.
4- Authors should explain that is the rational in the selection of the Mpro structure under PDB code 7KX5.
5- Authors state in lines 128-129 that “Finally, 17 compounds were selected based on the docking scores, MM/GBSA estimation and manually selection”, but fail to indicate the criteria for the manual selection.
6- Authors fail to appropriately show their results. Even if they present as supplementary information the PDB structures of their results they must show them comprehensively in the main text, they cannot rely on the fact that anyone who reads their work has the possibility to visualize the PDB structures. In lines 153-154 they state that “The binding conformations of these two ligands with the pocket were then investigated, respectively”, but there is no image of the 3D interaction of the ligand with the protein highlighting the amino acids that interact with the protein while showing the 3D structure of the latter. Figure 5 is not enough to back up the results as a 2D interaction diagram fails to give all the needed information. The PDB web page initially had 2D diagrams but have been removed favoring 3D interaction diagrams as the 2D diagrams either lacked critical information or where misleading.
7- The meaning of the results shown in Figure 4 should be explained more thoroughly. The 3D structures of the docking results should be shown (even on Supplementary information) so as to compare them to the resulting 3D structures from the MD. This way the RMSD would make more sense.
8- MM/GBSA results would be improved greatly if the per-residue decomposition energy was shown for the interaction. This would shed light on the importance of certain amino acids over others.
9- Authors state in lines 266-267 that “And final candidates were selected based on the docking scores, binding free energy estimation and manually selection” but fail to explain the criteria for that selection.
10- In line 303 authors state that they have carried out 300 ns, but in line 159 state that they performed 500ns. This should be clarified.
11- In all, all the images in the text are of very poor quality which makes the text in the images impossible to read. Authors should improve that.
12- There are citations missing in lines 58, 108, 114, 125, 215, 235, 236, 241, 242, 249, 254, 263, 295, 296 and 306.
The text should be thoroughly cheked for spelling and grammar mistakes.
Author Response
The authors appreciate the comments from the reviewer, which can greatly improve the quality of the manuscript. Here is the response to the comments.
The introduction is very thorough in the explanation of the life cycle of the virus, but could be improved by the addition of a schematic representation of that cycle.
A schematic life cycle was added.
The introduction fails to correctly explain the catalytic mechanism of the Mpro. Line 88 in page 2 states that “could covalently bind with the vital catalyze residue His41”. Residue His41 does not carry out the nucleophilic attack on the substrate or covalent inhibitors. His41 activates Cys145 and it is Cys145 that carries out the nucleophilic attack.
The authors thank the reviewer catch this point, which has been corrected.
The introduction fails to show the main point of interest of this article which is the structure of the Mpro and its interaction with bound inhibitors. This is crucial, as further along the text authors state in line 166 that they “identified two novel scaffolds” and in lines 249 “The binding pocket was identified”; but they have not provided any information whatsoever on the structure of the ligand-binding site of the known inhibitors, so the reader must take their word for it. Authors should at least use the crystal structure of Mpro bound to a ligand that they have used as starting structure for the computational experiments to explain the 3D structure of Mpro, the ligand-binding site and the key ligand-receptor interactions to be able to support their results further along the text.
The introduction of the crystal structure of Mpro and the ligand-receptor interactions have been added.
Authors should explain that is the rational in the selection of the Mpro structure under PDB code 7KX5.
Few cocrystal structures of Mpro crystal structure were reported when this work started. Most of them are covalent inhibitors and the crystal structure coded 7KX5 is the best choice which bound non-covalent inhibitor. The explanation has been added.
Authors state in lines 128-129 that “Finally, 17 compounds were selected based on the docking scores, MM/GBSA estimation and manually selection”, but fail to indicate the criteria for the manual selection.
The criteria for the manual selection have been added.
Authors fail to appropriately show their results. Even if they present as supplementary information the PDB structures of their results they must show them comprehensively in the main text, they cannot rely on the fact that anyone who reads their work has the possibility to visualize the PDB structures. In lines 153-154 they state that “The binding conformations of these two ligands with the pocket were then investigated, respectively”, but there is no image of the 3D interaction of the ligand with the protein highlighting the amino acids that interact with the protein while showing the 3D structure of the latter. Figure 5 is not enough to back up the results as a 2D interaction diagram fails to give all the needed information. The PDB web page initially had 2D diagrams but have been removed favoring 3D interaction diagrams as the 2D diagrams either lacked critical information or where misleading.
The 3D interaction images have been added.
The meaning of the results shown in Figure 4 should be explained more thoroughly. The 3D structures of the docking results should be shown (even on Supplementary information) so as to compare them to the resulting 3D structures from the MD. This way the RMSD would make more sense.
The 3D interaction images have been added.
MM/GBSA results would be improved greatly if the per-residue decomposition energy was shown for the interaction. This would shed light on the importance of certain amino acids over others.
Detailed MM/GBSA results have been provided as supplementary information.
Authors state in lines 266-267 that “And final candidates were selected based on the docking scores, binding free energy estimation and manually selection” but fail to explain the criteria for that selection.
The criteria for the manual selection have been added.
In line 303 authors state that they have carried out 300 ns, but in line 159 state that they performed 500ns. This should be clarified.
The authors thank the reviewer catch this point, which has been corrected.
In all, all the images in the text are of very poor quality which makes the text in the images impossible to read. Authors should improve that.
The high-resolution images have been uploaded as supplementary information.
There are citations missing in lines 58, 108, 114, 125, 215, 235, 236, 241, 242, 249, 254, 263, 295, 296 and 306.
More references have been added.
Reviewer 3 Report
This work aims at proposing with the aid of computer screening new compounds that might be efficient inhibitors of the so-called main protease (MPro), an important enzyme produced by SARS-Cov-2. The authors used a 3-step molecular docking against binding pocket as observed in a cocrystal structure of MPro with its inhibitor to identify 17 structures out of more than 3.5 million ones present in two commercial databases. A further selection was carried out based on the inhibitor activity of the identified compounds that were purchased and their inhibitory activity was investigated with a FRET assay. This approach ultimately led to two hits. The paper probably deserves publication. However, several points have to be addressed to make it clearer.
1) What is the meaning of the docking score shown in Table 1? How this number is calculated? What is its physical meaning?
2) Generally speaking, one may observe a competitive or allosteric inhibition. The paper suggests that the authors study competitive inhibition since the binding pocket structure of Mpro was used for molecular docking. However, one cannot exclude allosteric inhibition among the studied group of 17 compounds. The authors should prove that the mode of inhibition does not change within the studied group and that it is indeed a competitive inhibition.
3) What is the meaning of column 3 (%inhibition@100μM 3 samples average) in Table 2. Does it mean that data in column 2 were obtained in a single experiment (no repetition)? The lack of standard deviation for data gathered in column 2 does indicate that the experiments were not replicated. There is no point to show the experimental data without standard deviations. This has to be corrected.
4) I do not see a good reason to show the 2.4. “Binding free energy (MM/GBSA) calculation” section. It does not contribute anything to the discussion. Remove it.
5) In section 2.5 concerning the ADMET profile most of the symbols related to physicochemical properties are not explained. Moreover, in the referee’s opinion, there is no substitute for the ADMET profile than preclinical (animal) studies. So the value of data shown in section 2.5 is lower than those described in the remaining part of the article. Therefore, I suggest moving them to supplementary data.
6) I also suggest removing a lengthy description of the replication of SARS-CoV-2 (a paragraph starting with “The replication of SARS-CoV-2 can be divided into several steps,…” since it does not offer any valuable input to the paper. Most of the paragraph has nothing to do with Mpro and its inhibition.
Minor editing of English is required. For instance, "homodimer, consisted by two monomers"
Author Response
The authors appreciate the comments from the reviewer, which can greatly improve the quality of the manuscript. Here is the response to the comments.
What is the meaning of the docking score shown in Table 1? How this number is calculated? What is its physical meaning?
The virtual screening in this manuscript using Glide, which a module of Schrödinger suites. Glide uses the Emodel scoring function to select between protein-ligand complexes of a given ligand and the GlideScore function to rank-order compounds to separate compounds that bind strongly (actives) from those that don’t (inactives). The Emodel scoring function is primarily defined by the protein-ligand coulomb-vdW energy with a small contribution from GlideScore. GlideScore is an empirical scoring function designed to maximize separation of compounds with strong binding affinity from those with little to no binding ability. As an empirical scoring function, it is comprised of terms that account for the physics of the binding process including a lipophilic-lipophilic term, hydrogen bond terms, a rotatable bond penalty, and contributions from protein-ligand coulomb-vdW energies. In addition to these terms, GlideScore includes terms to account for hydrophobic enclosure3, which is the displacement of water molecules by a ligand from areas with many proximal lipophilic protein atoms. Particularly beneficial to binding is the formation of one or more protein-ligand hydrogen bonds within regions of hydrophobic enclosure.
The detailed scoring function introduction can be founded at:
https://www.schrodinger.com/science-articles/docking-and-scoring.
Glide: A New Approach for Rapid, Accurate Docking and Scoring. 1. Method and Assessment of Docking Accuracy, Friesner et al, J. Med. Chem., 2004, 47, 1739–1749.
Glide: A New Approach for Rapid, Accurate Docking and Scoring. 2. Enrichment Factors in Database Screening, Halgren et al, J. Med. Chem., 2004, 47, 1750–1759
Generally speaking, one may observe a competitive or allosteric inhibition. The paper suggests that the authors study competitive inhibition since the binding pocket structure of Mpro was used for molecular docking. However, one cannot exclude allosteric inhibition among the studied group of 17 compounds. The authors should prove that the mode of inhibition does not change within the studied group and that it is indeed a competitive inhibition.
The authors acknowledge the suggestion of the reviewer. Until now, few Mpro allosteric inhibitors have been reported. The potential allosteric binding site of Mpro is still unclear. Further, the compounds showed in this manuscript showed moderate inhibition activity to Mpro, which showed less interest for further detailed investigation.
What is the meaning of column 3 (%inhibition@100μM 3 samples average) in Table 2. Does it mean that data in column 2 were obtained in a single experiment (no repetition)? The lack of standard deviation for data gathered in column 2 does indicate that the experiments were not replicated. There is no point to show the experimental data without standard deviations. This has to be corrected.
The authors acknowledge the suggestion of the reviewer. Actually, the enzyme inhibition assay of all molecules was performed by a high throughput FRET screening together with a compound library. And those that showed acceptable inhibition potency were further validated to get the inhibition rate with 3 time average.
I do not see a good reason to show the 2.4. “Binding free energy (MM/GBSA) calculation” section. It does not contribute anything to the discussion. Remove it.
The authors acknowledge the suggestion of the reviewer. The MM/GBSA calculation is one of the most popular way to estimate the binding affinity of ligands. Detailed MM/GBSA calculation data have been provided as supplementary information.
In section 2.5 concerning the ADMET profile most of the symbols related to physicochemical properties are not explained. Moreover, in the referee’s opinion, there is no substitute for the ADMET profile than preclinical (animal) studies. So the value of data shown in section 2.5 is lower than those described in the remaining part of the article. Therefore, I suggest moving them to supplementary data.
The authors acknowledge the suggestion of the reviewer. ADMET profile investigation is one of the most procedures during drug development. However, it’s expensive and time-consuming. So, many researchers use in-silico ADMET estimation to evaluate the lead compounds before preclinical (animal) studies.
I also suggest removing a lengthy description of the replication of SARS-CoV-2 (a paragraph starting with “The replication of SARS-CoV-2 can be divided into several steps,…” since it does not offer any valuable input to the paper. Most of the paragraph has nothing to do with Mpro and its inhibition.
The authors acknowledge the suggestion of the reviewer. The introduction of SARS-CoV-2 life cycle can help understanding the importance of Mpro. Further, the crystal structure of Mpro and inhibitor complex were also illustrated, which could show the vital role of Mpro and the meaning of Mpro inhibitor development.
Reviewer 4 Report
This manuscript describes the “in silico” search for inhibitors against the SARS-CoV-2 main protease. The authors performed docking study, binding free energy predictions with MM-GB/SA, inhibitory activity assay with FRET, and ADMET predictions. For in-silico studies, the authors used mainly commercially available tools. They found out two potential strong inhibitors, AF-399/407 13777 and AI-942/42301830. The topic treated by authors is interesting and significant but I feel that the authors should reconsider the following points before publication.
The title of this manuscript is “In silico discovery..” but the criterion to determine a probably strong inhibitor first is experimental inhibitory activity assay. They show the docking scores predicted binding free energies in Table 1 but these data do not indicate that proposed two compounds are strongest inhibitors. MD simulations are just indicating the stabilities of binding poses of these two compounds. If the authors insist the stabilities of binding poses of proposed compounds, they should perform the same MD simulations for other compounds. If predicted binding free energy is also used for the determination, AN-329/15538195 should be also included in the discussion. If the authors keep the experimental assay as the criterion of inhibitor selections, I think that the authors should make further experiments to determine Ki or IC50 values and so on.
Minor points;
1. There is a sentence “Several block- 107 buster drugs (Glivec, etc) were developed using CADD.” in page 3, line 107 and 108. The citation for this sentence should be done.
2. The more detailed caption of Figure 4 should be required. What are the red and blue lines? What is the green bars?
Author Response
The authors appreciate the comments from the reviewer, which can greatly improve the quality of the manuscript. Here is the response to the comments.
There is a sentence “Several block- 107 buster drugs (Glivec, etc) were developed using CADD.” in page 3, line 107 and 108. The citation for this sentence should be done.
More references have been added.
The more detailed caption of Figure 4 should be required. What are the red and blue lines? What is the green bars?
The figure caption has been added.
Round 2
Reviewer 1 Report
Improvements were made.
Improvements were made.
Author Response
The authors appreciate the comments from the reviewer, which are a great improvement to the manuscript's quality.
Reviewer 3 Report
The authors improved their presentatioin and addressed my doubts.
Author Response

(The authors gave the same response as above.)

Reviewer 4 Report
The authors have replied for only my minor points.
The authors should reconsider the following points, and revise the manuscript or at least give some comments.
"The title of this manuscript is “In silico discovery..” but the criterion to determine a probably strong inhibitor first is experimental inhibitory activity assay. They show the docking scores predicted binding free energies in Table 1 but these data do not indicate that proposed two compounds are strongest inhibitors. MD simulations are just indicating the stabilities of binding poses of these two compounds. If the authors insist the stabilities of binding poses of proposed compounds, they should perform the same MD simulations for other compounds. If predicted binding free energy is also used for the determination, AN-329/15538195 should be also included in the discussion. If the authors keep the experimental assay as the criterion of inhibitor selections, I think that the authors should make further experiments to determine Ki or IC50 values and so on."
Author Response
The authors apologize for omitting reviewers' comments. It’s appreciated by the authors to the comments of the reviewer. Here is the response to the comments.
"The title of this manuscript is “In silico discovery..” but the criterion to determine a probably strong inhibitor first is experimental inhibitory activity assay. They show the docking scores predicted binding free energies in Table 1 but these data do not indicate that proposed two compounds are strongest inhibitors. MD simulations are just indicating the stabilities of binding poses of these two compounds. If the authors insist the stabilities of binding poses of proposed compounds, they should perform the same MD simulations for other compounds. If predicted binding free energy is also used for the determination, AN-329/15538195 should be also included in the discussion. If the authors keep the experimental assay as the criterion of inhibitor selections, I think that the authors should make further experiments to determine Ki or IC50 values and so on."
The authors acknowledge the suggestion of the reviewer. Actually, the enzyme inhibition assay of all selected molecules was performed by a high throughput FRET screening together with a compound library. And those that showed acceptable inhibition potency were further validated to get the inhibition rate with 3 time average. AN-329/15538195 showed interesting inhibition potency to Mpro. However, comparing AF-399/40713777 and AI-942/42301830, AN-329/15538195 showed less potency in the FRET inhibition assay. Thus, the further study focuses on AF-399/40713777 and AI-942/42301830. And the following MD simulation mainly focuses on the investigation of the possible binding poses of AF-399/40713777 and AI-942/42301830, which are the most interesting compounds identified in this screening. Figure 5 has been updated.